# Dynamic Model of Collaboration in Multi-Agent System Based on Evolutionary Game Theory

Zhuozhuo Gou and Yansong Deng *

Key Laboratory of Electronic Information of State Ethnic Affairs Commission, Southwest Minzu University, Chengdu 610041, China; gouzhuozhuo0778@163.com
* Correspondence: dengyansong888@163.com; Tel.: +86-18980609123

**Abstract:** Multi-agent collaboration is greatly important in order to reduce the frequency of errors in message communication and enhance the consistency of exchanging information. This study explores the process of evolutionary decision and stable strategies among multi-agent systems, including followers, leaders, and loners, involved in collaboration based on evolutionary game theory (EGT). The main elements that affected the strategies are discussed, and a 3D evolution model is established. The evolutionary stability strategy (ESS) and stable conditions were analyzed subsequently. Numerical simulation results were obtained through MATLAB simulation, and they manifested that leaders play an important role in exchanging information with other agents, accepting agents' state information, and sending messages to agents. Then, with the positivity of receiving and feeding back messages for followers, implementing message communication is profitable for the system, and the high positivity can accelerate the exchange of information. At the behavior level, reducing costs can strengthen the punishment of impeding the exchange of information and improve the positivity of collaboration to facilitate the evolutionary convergence toward the ideal state. Finally, the EGT results revealed that the possibility of collaboration between loners and others is improved, and the rewards are increased, thereby promoting the implementation of message communication that encourages leaders to send all messages, improve the feedback positivity of followers, and reduce the hindering degree of loners.

**Keywords:** collaboration; consistency; evolutionary stability strategies; multi-agent; evolutionary game

## 1. Introduction

A multi-agent system is an important branch of distributed artificial intelligence, and several independent agents are adopted to achieve common goals in this system. These agents have an autonomous ability to coordinate with each other. In multi-agent systems, the research on the system's collaboration control has mainly involved tracking [1–4], formation [5–7], swarm [8–10], rendezvous [11], distributed filtering [12], and consistency [13,14]. Collaboration consistency elucidates that the state of all agents tends toward the same tendency, and it illustrates the rule of interacting and transmitting information when agents cooperate with other agents; additionally, it describes the process of information exchange between each agent and other agents. When agents are able to deal with various unpredictable situations and suddenly variable environments, the effectiveness of collaboration is reflected in reaching consensus on goals as the environment changes. Therefore, the agreement of multi-agents to achieve common goals is a primary condition for collaborative control.

In previous works, collaboration consistency was first applied to solve the problem of fusion under uncertain information in multi-sensors in 1974 [15]. In the subsequent few years, Borkar et al. [16,17] studied synchronous asymptotic consistency, which was adopted to investigate the decision of a distributed system in the field of control theory. In 1995 [18], Vicsek et al. proposed a classical model, that is, the dispersion system of

multi-agents moving in a plane to simulate the phenomenon of particles presenting coherent behavior. Through the introduction of graph theory and matrix theory in 2003 [19], Jadbabaie explained the theory of consistency and found that the sets of agents' neighbors varied over time in the system. Subsequently, R. Olfati et al. [20–22] described a framework of theory to figure out a consistency problem for dynamic systems. In 2010 [23], researchers observed the problem of consistency and synchronization of multi-agent systems in complex networks. Over the last few decades, researchers have explored this collaboration from different aspects. Some researchers focused on controlling groups of autonomous mobile vehicles to implement concentrated and decentralized collaboration control [24]. Two basic controllers of leader–follower control were proposed to allow the followers to maintain a relative position and avoid collisions in front of the obstacles. Different from other studies on leader–follower approaches, a recent article has suggested that the orientation deviations of leader–followers be explicitly expressed in the model to successfully solve collaboration controls when the agents move backward [25].

In previous studies, the agents' consistency has been investigated in simple integrators, whereas agents are complex in practical engineering applications. In addition, it is not in line with the conditions of actual applications under complex and changeable environments.

In recent years, with the continuous efforts of researchers, the consistency of static and dynamic networks has been adopted in various fields to satisfy practical applications. In terms of the consistency of collaboration in formation control, leadership–follow strategy [26,27] indicates that some agents are leaders and others are followers who track the position and direction of the leaders at a certain distance. Some researchers [28] investigated the leader–follower formation control model based on uncertain nonholonomic-wheeled mobile robots. They also expressed that the leaders' signal can be smooth, feasible, or nonfeasible. Adopting the estimated states of a leader, they transform formation errors into external oscillator states in an augmented system that presents additional control parameters that overcome actuation difficulties and reduce formation errors. One article [29] manifested the problem of formation control based on the leader–follower model in 3D space, which explores the persistent excitation of the desired formation to achieve the exponential stabilization of actual formation in terms of shape and scale. In general, designing these controllers to realize and describe the collaboration of agents is easy. However, considering the operating capability of different agents is difficult. Ignoring the perspective of global programming limits the effectiveness of the collaboration, which can be resolved in a distributed coordination approach. Then, for the consistency of collaboration, researchers [30,31] have investigated swarming motility in various networks. Tanner and Jadbabaie [32] proved the stability of swarm control and proposed a new protocol of consistency to analyze the stable properties of mobile agents and stabilize their inter-agent distances, adopting the rules of decentralized and nearest-neighbor interaction with exchanging information. The discontinuities of control laws are introduced via these changes. Nonsmooth analysis is used to accommodate arbitrary switching based on the network of interactions. The main result shows that regardless of switching, a common velocity vector is guaranteed to reach a convergence state when the network remains connected all along. Moreover, the collaboration based on the evolutionary game is analyzed thoroughly in small-world and scale-free networks [33–35]. Meanwhile, researchers described the consistency of fixed and switched topology in a multi-agent system [36–38], where each agent is a universal linear dynamic system and a linear model of nonlinear networks. Thus, a unified framework for complex networks is set up.

In summary, the existing studies only consider the interactions of leaders–followers. In reality, environmental factors play an indispensable role in the exchange of information. Therefore, the interactions of three stakeholders involved in the collaboration should be investigated.

Hence, in response to this discussion, the process of evolutionary decision and stable strategies among three stakeholders, such as followers, leaders, and loners, involved in

the collaboration of a multi-agent system based on evolutionary game theory (EGT) is demonstrated. The main elements that affected the strategies of the agents are discussed, and the 3D replicator evolution equation is established to obtain the evolutionary stability strategy (ESS). Stable conditions are acquired through the theory of Lyapunov stability. The reasonability of the proposed mechanism is confirmed by simulation experiments. This research may help the agents to make optimal decisions and may provide theoretical guidance to agents to implement collaboration and adapt to complex environments. The contributions of this study are presented as follows:

(1) We establish a tripartite dynamic evolution model of followers, leaders, and loners for collaboration. Different from the previous game model, which involved only two stakeholders of leadership–followers, this model investigates the influence of factors and the exchange of information among the three stakeholders effectively.

(2) The main parameters of the strategies are involved in feedback, sending, and receiving messages for three parties, namely, the followers, leaders, and loners, respectively; these parameters are analyzed in the simulation discussion. Moreover, other influential factors, including the degree of positivity and the possibility of interaction, are discussed in the game model. We figure out the evolutionary stable strategies (ESS) of agents under different stability conditions and scenarios.

(3) The simulation results indicate that when the possibility of collaboration between loners and others is improved and when the rewards are increased, the implementation of message communication can be promoted to encourage leaders to send all messages, improve the positivity of feedback for followers, and reduce the hindering degree of loners.

(4) Finally, conclusions are obtained and policy implementation is put forward to offer suggestive guidance of actual application.

The remainder of this paper is presented as follows: We describe the evolution of the game model in Section 2. Then, Section 3 illustrates the equilibrium points and stability analysis. In Section 4, the simulation results and discussion are confirmed. Finally, our conclusions and policy enlightenment are figured out in Section 5.

## 2. Model

In this section, the dynamic collaboration model based on evolutionary game theory is proposed. Then, the payoff matrix of the agents is obtained according to the parameters of the agents' behavior. In addition, the tripartite replication dynamic equation is derived.

### 2.1. Descriptions and Notes of the Parameters in a Multi-Agent System

In a multi-agent system, each agent can work by itself or in an environment and interact with other agents. Thus, mutually independent agents deal with complex problems in the coordinated approach to achieve a common goal. However, agents may be disturbed by external factors in a hostile environment when completing tasks, thereby resulting in the failure of normal communication. We assume that the interferential factors are described by the loners' behavior. As shown in Figure 1, in the multi-agent system model, different agents perform their own various tasks. Initially, leaders send messages to followers and loners. Then, the followers decide whether to provide feedback to the leaders after receiving messages while sending messages to the loners. Subsequently, the loners can select whether or not to receive messages. If the loners receive messages, then the destructive power of the loner decreases when they communicate with each other. Otherwise, the destructive power of the loner increases in the exchange of information in a changing environment.

All agents have the right to select their own decisions in the communication process. Therefore, the set of strategies for followers is {feedback, not feedback}. Regardless of sending all or partial messages to the leaders, the followers receive messages and obtain payoffs. Then, the followers decide whether or not the processed information is fed back to the leaders. They can obtain rewards when feeding back to the leaders. Otherwise, they obtain nothing and do not involve cost.

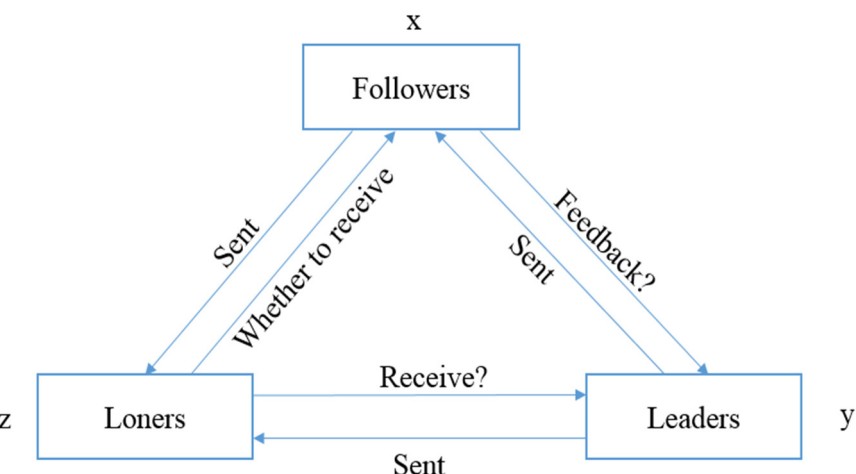

**Figure 1.** Multi-agent system model.

For leaders, the set of strategies is {all messages, partial messages}. They obtain payoffs $P_{L1}$ with the cost of $C_{L1}$ when sending all messages. When sending partial messages, they obtain $P_{L2}$ payoffs under the cost of $C_{L2}$. Leaders can gain rewards $R_L$ as the feedback messages are received. We assume that $P_{L1}$ is greater than $P_{L2}$, and $C_{L1}$ is more than $C_{L2}$.

In terms of loners, the strategy set is {receive, not receive}. $P_{z1}$ represents the payoff of loners receiving messages from followers with the cost of $C_{z1}$. The loners obtain payoffs $P_{z2}$ when messages from the leaders are received under the cost of $C_{z2}$. $R$ represents the payoff of unsuccessfully receiving messages. Interactive rewards $I_f$ and $I_L$ can be obtained as loners interact with the followers and leaders, respectively.

Other parameters and notes are described as follows: $P_{f1}$ represents the payoffs of receiving messages for followers at the cost of $C_{f1}$. $P_{f2}$ indicates the payoffs that followers obtain as they send messages to loners with the cost of $C_{f2}$. We adopt parameters $\alpha$ and $\beta$ to describe the degree of positivity for feedback and receiving, respectively, to define the positivity. $p$ represents the probability of successful sending. $\gamma$ is the interactive possibility of agents communicating with others. For interactive rewards, we stipulate that the value of $I_f$ is the same as that of $I_L$. Specific parameters and notes can be represented in Table 1.

**Table 1.** Parameter descriptions and notes.

| Parameters | Descriptions | Notes |
|---|---|---|
| $P_{f1}, C_{f1}$ | Profits and costs of followers receiving messages, respectively. | $P_{f1} > 0, C_{f1} > 0$ |
| $P_{f2}, C_{f2}$ | Profits and costs of sending messages to loners, respectively. | $P_{f2} > 0, C_{f2} > 0$ |
| $R_f, C_f$ | Rewards and costs of followers sending feedback messages to leaders. | $R_f > 0, C_f > 0$ |
| $\alpha$ | Positive degree of feedback to leaders. | $\alpha > 0$ |
| $\beta$ | Positive degree of reception. | $\beta > 0$ |
| $P_{L1}, C_{L1}$ | Profits and costs of leaders sending all messages, respectively. | $P_{L1} > 0, C_{L1} > 0$ |
| $P_{L2}, C_{L2}$ | Profits and costs of leaders sending partial messages, respectively. | $P_{L1} > P_{L2} > 0, C_{L1} > C_{L2} > 0$ |
| $R_L, C_L$ | Rewards and costs of receiving messages from followers, respectively. | $R_L > 0, C_L > 0$ |
| $p$ | Probability of sending messages successfully. | $0 < p < 1$ |
| $\lambda$ | $\lambda = 1$ indicates all messages, $0 < \lambda < 1$ represents partial messages. | |
| $P_{z1}, C_{z1}$ | Profits and costs of loners receiving messages from followers. | $P_{z1} > 0, C_{z1} > 0$ |
| $P_{z2}, C_{z2}$ | Profits and costs of receiving messages from leaders, respectively. | $P_{z2} > 0, C_{z2} > 0$ |
| $R$ | Profits of receiving messages unsuccessfully. | $R > 0$ |
| $\gamma$ | Possibility of interaction when loners receive messages successfully. | $0 < \gamma < 1$ |
| $I_f, I_L$ | Rewards of interacting with followers and leaders respectively. | $I_f > 0, I_L > 0$ |

### 2.2. Payoff Matrix of Agents

The proportion of strategies in the agents' population can be denoted as follows. $x$ $(0 \leq x \leq 1)$ is the probability of followers feeding back messages. On the contrary, the probability of nonfeedback is $1 - x$. For leaders, $y$ $(y = 1)$ represents the proportion of sending all messages. $0 < y < 1$ indicates the proportion of sending partial messages. In terms of loners, assuming that the probability of receiving messages is $z$, $1 - z$ denotes the proportion of nonreceiving messages. The corresponding payoff matrix is shown in Table 2.

**Table 2.** Payoff matrix of followers, leaders, and loners.

| | **All ($y = 1$)** | | **Partial ($0 < y < 1$)** | |
| --- | --- | --- | --- | --- |
| | **Receive ($z$)** | **Not receive ($1-z$)** | **Receive ($z$)** | **Not receive ($1-z$)** |
| Feedback ($x$) | $(Fop_1, Lep_1, Lop_1)$ | $(Fop_2, Lep_2, Lop_2)$ | $(Fop_3, Lep_3, Lop_3)$ | $(Fop_4, Lep_4, Lop_4)$ |
| Not feedback ($1 - x$) | $(Fop_5, Lep_5, Lop_5)$ | $(Fop_6, Lep_6, Lop_6)$ | $(Fop_7, Lep_7, Lop_7)$ | $(Fop_8, Lep_8, Lop_8)$ |

In accordance with the different strategies decided by agents, the corresponding payoffs can be obtained. *Fop*, *Lep*, and *Lop* represent the payoffs of followers, leaders, and loners, respectively. The specific expression is shown in the following equations:

$$Fop_1 = \beta P_{f1} + \beta P_{f2} - C_{f1} - C_{f2} + \alpha R_f - C_f \tag{1}$$

$$Lep_1 = pP_{L1} - C_{L1} + \beta R_L - C_L \tag{2}$$

$$Lop_1 = \beta P_{z1} + \beta P_{z2} - C_{z1} - C_{z2} + \gamma \left( I_f + I_L \right) \tag{3}$$

$$Lep_2 = pP_{L1} - C_{L1} + \beta R_L - C_L \tag{4}$$

$$Lep_2 = pP_{L1} - C_{L1} + \beta R_L - C_L \tag{5}$$

$$Lop_2 = R \tag{6}$$

$$Fop_3 = \lambda \left( \beta P_{f1} + \beta P_{f2} - C_{f1} - C_{f2} + \alpha R_f - C_f \right) \tag{7}$$

$$Lep_3 = pP_{L2} - C_{L2} + \lambda(\beta R_L - C_L) \tag{8}$$

$$Lop_3 = \lambda(\beta P_{z1} + \beta P_{z2} - C_{z1} - C_{z2}) + \gamma \left( I_f + I_L \right) \tag{9}$$

$$Fop_4 = \lambda \left( \beta P_{f1} + \beta P_{f2} - C_{f1} - C_{f2} + \alpha R_f - C_f \right) \tag{10}$$

$$Lep_4 = pP_{L2} - C_{L2} + \lambda(\beta R_L - C_L) \tag{11}$$

$$Lop_4 = R \tag{12}$$

$$Fop_5 = \beta P_{f1} + \beta P_{f2} - C_{f1} - C_{f2} \tag{13}$$

$$Lep_5 = pP_{L1} - C_{L1} \tag{14}$$

$$Lop_5 = \beta P_{z1} + \beta P_{z2} - C_{z1} - C_{z2} + \gamma \left( I_f + I_L \right) \tag{15}$$

$$Fop_6 = \beta P_{f1} + \beta P_{f2} - C_{f1} - C_{f2} \tag{16}$$

$$Lep_6 = pP_{L1} - C_{L1} \tag{17}$$

$$Lop_6 = R \tag{18}$$

$$Fop_7 = \lambda \left( \beta P_{f1} + \beta P_{f2} - C_{f1} - C_{f2} \right) \tag{19}$$

$$Lep_7 = pP_{L2} - C_{L2} \tag{20}$$

$$Lop_7 = \lambda(\beta P_{z1} + \beta P_{z2} - C_{z1} - C_{z2}) + \gamma\left(I_f + I_L\right) \tag{21}$$

$$Fop_8 = \lambda\left(\beta P_{f1} + \beta P_{f2} - C_{f1} - C_{f2}\right) \tag{22}$$

$$Lep_8 = pP_{L2} - C_{L2} \tag{23}$$

$$Lop_8 = R \tag{24}$$

### 2.3. Replication Dynamic Equation of Agents

The expected revenue can be obtained according to the payoff matrix. Let $E_x$ represent the expected payoffs of followers when they feedback messages. Similarly, $E_{1-x}$ represents the expected payoffs of nonfeedback, as shown in the equation where $\overline{E}_x$ is the average of expected payoffs for followers. $E_x$, $E_{1-x}$, and $\overline{E}_x$ can be shown as follows:

$$E_x = yzFop_1 + y(1-z)Fop_2 + (1-y)zFop_3 + (1-y)(1-z)Fop_4 = [\lambda + y(1-\lambda)Fop_1] \tag{25}$$

$$E_{1-x} = yzFop_5 + y(1-z)Fop_6 + (1-y)zFop_7 + (1-y)(1-z)Fop_8 = [\lambda + y(1-\lambda)Fop_5] \tag{26}$$

$$\overline{E}_x = xE_x + (1-x)E_{1-x} \tag{27}$$

Hence, we can obtain the dynamic equation of followers as follows:

$$F_x = \frac{dx}{dt} = x(E_x - \overline{E}_x) = x(1-x)[\lambda + y(1-\lambda)]\left(\alpha R_f - C_f\right) \tag{28}$$

They obtain the expected payoffs $E_y$ when the leaders send all messages. Similarly, $E_{1-y}$ is the expected payoffs of sending partial messages, as shown in the equation where $\overline{E}_y$ is the average of the expected payoffs.

$$E_y = xzLep_1 + x(1-z)Lep_2 + (1-x)zLep_5 + (1-x)(1-z)Lep_6 = pP_{L1} - C_{L1} + x(\beta R_L - C_L) \tag{29}$$

$$E_{1-y} = xzLep_3 + x(1-z)Lep_4 + (1-x)zLep_7 + (1-x)(1-z)Lep_8 = pP_{L2} - C_{L2} + x\lambda(\beta R_L - C_L) \tag{30}$$

$$\overline{E}_y = yE_y + (1-y)E_{1-y} \tag{31}$$

Therefore, the dynamic equation of the leaders can be expressed as follows:

$$F_y = \frac{dy}{dt} = y(E_y - \overline{E}_y) = y(1-y)[p(P_{L1} - P_{L2}) + (C_{L2} - C_{L1}) + x(\beta R_L - C_L)(1-\lambda)] \tag{32}$$

As loners receive messages, they obtain the expected payoffs $E_z$. Similarly, $E_{1-z}$ indicates the expected payoffs of nonreceiving, as shown in the equation where $\overline{E}_z$ is the average of the expected payoffs.

$$E_z = xyLop_1 + y(1-x)Lop_5 + (1-y)xLop_3 + (1-x)(1-y)Lop_7 = Lop_3 + y(1-\lambda)(\beta P_{z1} - C_{z1} + \beta P_{z2} - C_{z2}) \tag{33}$$

$$E_{1-z} = xyLop_2 + y(1-x)Lop_6 + (1-y)xLop_4 + (1-x)(1-y)Lop_8 = R \tag{34}$$

$$\overline{E}_z = zE_z + (1-z)E_{1-z} \tag{35}$$

The dynamic equation of loners can be expressed as follows:

$$F_z = \frac{dz}{dt} = z(E_z - \overline{E}_z) = z(1-z)\left\{[\beta(P_{z1} + P_{z2}) - (C_{z2} + C_{z1})][\lambda + y(1-\lambda)] + \gamma\left(I_f + I_L\right) - R\right\} \tag{36}$$

Finally, the 3D dynamic equations of the system are expressed by the replicated dynamic equations of the followers, leaders, and loners, as follows:

$$F_x = \frac{dx}{dt} = x(1-x)[\lambda + y(1-\lambda)](\alpha R_f - C_f) \tag{37}$$

$$F_y = \frac{dy}{dt} = y(1-y)[p(P_{L1} - P_{L2}) + (C_{L2} - C_{L1}) + x(\beta R_L - C_L)(1-\lambda)] \tag{38}$$

$$F_z = \frac{dz}{dt} = z(1-z)\{[\beta(P_{z1} + P_{z2}) - (C_{z2} + C_{z1})][\lambda + y(1-\lambda)] + \gamma(I_f + I_L) - R\} \tag{39}$$

### 3. Equilibrium Point and Stability Analysis

To simplify the calculation, complex formulas of these dynamic equations can be expressed by simple letters as follows:

$$a = \alpha R_f - C_f, b = \beta R_L - C_L, d = p(P_{L1} - P_{L2}) + (C_{L2} - C_{L1}), e = \beta(P_{z1} + P_{z2}) - (C_{z2} + C_{z1}), f = \gamma(I_f + I_L) - R \tag{40}$$

Hence, 3D dynamic equations can also be expressed as follows:

$$F_x = \frac{dx}{dt} = x(1-x)[\lambda + y(1-\lambda)]a \tag{41}$$

$$F_y = \frac{dy}{dt} = y(1-y)[d + xb(1-\lambda))] \tag{42}$$

$$F_z = \frac{dz}{dt} = z(1-z)\{e[\lambda + y(1-\lambda)] + f\} \tag{43}$$

**Theorem 1.** *To gain equilibrium points, let $F_x$, $F_y$, and $F_z$ should be equal to 0 in Equations (41)–(43). Under the condition of pure strategies, we can obtain eight equilibrium points $p_1(0,0,0)$, $p_2(0,0,1)$, $p_3(1,0,0)$, $p_4$ (1,1,0), $p_5$ (1,0,1), $p_6$ (0,1,0), $p_7$ (0,1,1), and $p_8$ (1,1,1). According to dynamic equations, the loners use two equilibrium points for pure strategies, such as $p_9$ ($\frac{d}{(\lambda-1)b}$, $\frac{\lambda}{\lambda-1}$,0) and $p_{10}(\frac{d}{(\lambda-1)b}, \frac{\lambda}{\lambda-1},1)$. Among them, $0 < \frac{d}{(\lambda-1)b} < 1$, $0 < \frac{\lambda}{\lambda-1} < 1$.*

**Proof of Theorem 1.** Substituting the value of $x = 0$ or 1, $y = 0$ or 1, $z = 0$ or 1 into Equations (41)–(43), equations $F_x$, $F_y$, and $F_z$ equal to 0 are satisfied. As a result, $p_1$, $p_2$, $p_3$, $p_4$, $p_5$, $p_6$, $p_7$, and $p_8$ are equilibrium points of the system model. As $z = 0$, $0 < x < 1$, $0 < y < 1$, if $\lambda + y(1-\lambda) = 0$ and $d + xb(1-\lambda) = 0$, that is, $x = \frac{d}{(\lambda-1)b}, y = \frac{\lambda}{\lambda-1}$ is plugged into Equations (41)–(43), where $F_x = 0, F_y = 0$ and $F_z = 0$ can be obtained. Therefore, $p_9$ and $p_{10}$ are also equilibrium points. The multi-agent system model has no mixed strategy equilibrium point. □

According to the method of Frideman [39,40], $x$ is an evolutionary stable strategy as $F(x) = 0$ and $F'(x) = 0$. Jacobian matrix analyses of the stability of the system should be adopted for convenient calculation. The Jacobian matrix for the system can be described as follows:

$$J = \begin{bmatrix} \frac{\partial F(x)}{\partial x} & \frac{\partial F(x)}{\partial y} & \frac{\partial F(x)}{\partial z} \\ \frac{\partial F(y)}{\partial x} & \frac{\partial F(y)}{\partial y} & \frac{\partial F(y)}{\partial z} \\ \frac{\partial F(z)}{\partial x} & \frac{\partial F(z)}{\partial y} & \frac{\partial F(z)}{\partial z} \end{bmatrix} \tag{44}$$

The specific matrix representation is shown in the following equation:

$$J = \begin{bmatrix} (1-2x)a[\lambda + y(1-\lambda)] & x(1-x)a(1-\lambda) & 0 \\ y(1-y)b(1-\lambda) & (1-2y)[d + bx(1-\lambda)] & 0 \\ 0 & z(1-z)[e(1-\lambda)] & (1-2z)\{e[\lambda + y(1-\lambda)] + f\} \end{bmatrix} \tag{45}$$

Initially, the equilibrium points are carried out into the Jacobian matrix to obtain the eigenvalues. Then, whether the equilibrium points are stable or not is judged according to

the eigenvalues and limiting conditions; the results are shown in Table 3. Specifically, the eigenvalue $A_1$ of $p_9$ and $p_{10}$ is shown as follows:

$$A_1 = \left\{ \frac{b\lambda}{\lambda - 1} \left( -\frac{da}{b} \right) \left[ 1 - \frac{d}{(\lambda - 1)b} \right] \right\}^{\frac{1}{2}} \tag{46}$$

**Table 3.** Stability analysis of equilibrium points.

| Equilibrium Points | Eigenvalues $\lambda_1$ | Eigenvalues $\lambda_2$ | Eigenvalues $\lambda_3$ | Stability Condition |
|---|---|---|---|---|
| $p_1(0,0,0)$ | $a\lambda.$ | $d$ | $e\lambda + f$ | $a\lambda < 0, d < 0, e\lambda + f < 0$ |
| $p_2(0,0,1)$ | $a\lambda$ | $d$ | $-(e\lambda + f)$ | $a\lambda < 0, d< 0, e\lambda + f >0$ |
| $p_3(1,0,0)$ | $-a\lambda$ | $d + b(1 - \lambda)$ | $e\lambda + f$ | $a\lambda > 0, d + b(1 - \lambda) < 0, e\lambda + f < 0$ |
| $p_4(1,1,0)$ | $-a$ | $-[d + b(1 - \lambda)]$ | $e + f$ | $a > 0, d + b(1 - \lambda) > 0, e + f < 0$ |
| $p_5(1,0,1)$ | $-a\lambda$ | $d + b(1 - \lambda)$ | $-(e\lambda + f)$ | $a\lambda > 0, d + b(1 - \lambda)< 0, e\lambda + f >0$ |
| $p_6(0,1,0)$ | $a$ | $-d$ | $e + f$ | $a< 0, d >0, e + f < 0$ |
| $p_7(0,1,1)$ | $a$ | $-d$ | $-(e + f)$ | $a< 0, d >0, e + f > 0$ |
| $p_8(1,1,1)$ | $-a$ | $-[d + b(1 - \lambda)]$ | $-(e + f)$ | $a > 0, d + b(1 - \lambda) > 0, e + f > 0$ |
| $p_9\left( \frac{d}{(\lambda-1)b}, \frac{\lambda}{\lambda-1}, 0 \right)$ | $A_1$ | $-A_1$ | $f$ | unstable |
| $p_{10}\left( \frac{d}{(\lambda-1)b}, \frac{\lambda}{\lambda-1}, 1 \right)$ | $A_1$ | $-A_1$ | $-f$ | unstable |

## 4. Simulation Results and Discussion

The replication dynamic equation (RD) and the evolutionary stable strategy (ESS) constitute the core concepts of evolutionary game theory. They describe the dynamic convergence process to the steady-state of the evolutionary game. In RD, the time step of $t$ represents the derivative of the dynamic system of followers, leaders, and loners as follows:

$$\frac{dx(t)}{dt} = x(t)(1 - x(t))[\lambda + y(t)(1 - \lambda)]a \tag{47}$$

$$\frac{dy(t)}{dt} = y(t)(1 - y(t)[d + x(t)b(1 - \lambda))] \tag{48}$$

$$\frac{dz(t)}{dt} = z(t)(1 - z(t))\{e[\lambda + y(t)(1 - \lambda)] + f\} \tag{49}$$

Simulation experiments are carried out with different parameters to demonstrate the influence of the parameters on the convergence rate under the restricted condition of ESS. The length of time is set to 30.

### 4.1. Scenarios of Different Parameters with Constraint Conditions in the Equilibrium Points
#### 4.1.1. Scenario 1

In point $p_1(0,0,0)$, we can set the initial values of parameters $\lambda = 0.5$, $\alpha = 0.2$, $\beta = 0.2$, $\gamma = 0.1$, $p = 0.3$, $R_f = 20$, $C_f = 5$, $P_{L1} = 15$, $P_{L2} = 10$, $C_{L1} = 3$, $C_{L2} = 1$, $R_L = 20$, $C_L = 5$, $P_{z1} = 15$, $P_{z2} = 15$, $C_{z1} = 1, C_{z2} = 1$, $I_f = 15$, $I_L = 15$, $R = 10$. The number of leaders with high comprehensive ability and loners with weak cooperation ability is in the minority due to the agents' different abilities. At the initial time of dynamic evolution, that is, $t = 0$, we assume that the proportion of followers $x$ equals 0.5, the proportion of leaders $y$ equals 0.3, and the proportion of loners $z$ equals 0.2. The evolutionary results are denoted in Figure 2.

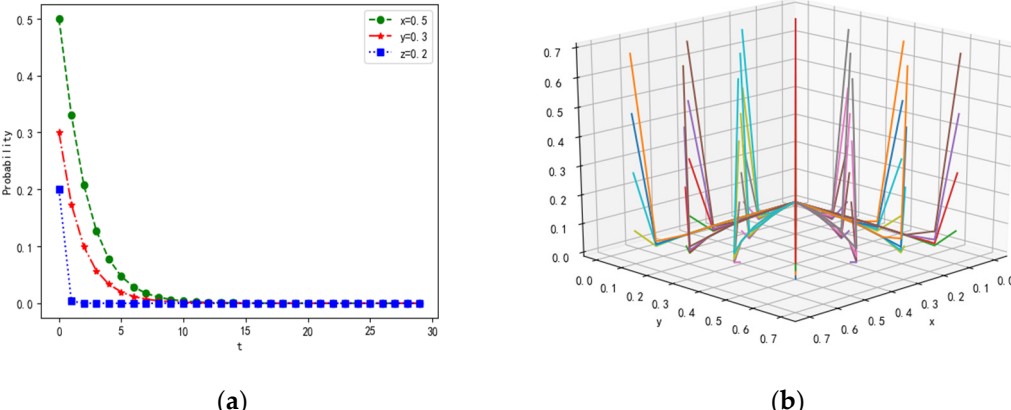

**Figure 2.** Evolutionary results of $p_1(0,0,0)$ are shown under a corresponding constraint condition, (**a**) represents the convergence of probability with time, and (**b**) shows the results of dynamic simulation in three dimensions.

For $a\lambda < 0$, $d < 0$, $e\lambda + f < 0$, that is, $\alpha R_f - C_f < 0$, $p(P_{L1} - P_{L2}) + (C_{L2} - C_{L1}) < 0$ and $\lambda[\beta(P_{z1} + P_{z2}) - (C_{z1} + C_{z2})] < R - \gamma(I_f + I_L)$, we find that the cost of tasks completed together is higher than the payoffs in the multi-agent system, thereby leading to the probability of feedback, sending, and receiving tending to zero over time. If the leaders do not send messages, then the followers and loners will not receive and feedback messages. This phenomenon is not conducive to the collaboration and interaction of the system.

### 4.1.2. Scenario 2

According to the stability conditions of $p_2(0,0,1)$, the parameter values are set as follows: $\lambda = 0.5$, $\alpha = 0.2$, $\beta = 0.2$, $\gamma = 0.3$, $p = 0.3$, $R_f = 20$, $C_f = 5$, $P_{L1} = 15$, $P_{L2} = 10$, $C_{L1} = 3$, $C_{L2} = 1$, $R_L = 20$, $C_L = 5$, $P_{z1} = 20$, $P_{z2} = 20$, $C_{z1} = 1$, $C_{z2} = 1$, $I_f = 15$, $I_L = 15$, $R = 10$. The probability of strategies remains unchanged. The simulation results are shown in Figure 3.

Under the condition of $\alpha R_f - C_f < 0$, $p(P_{L1} - P_{L2}) + (C_{L2} - C_{L1}) < 0$ and $\lambda[\beta(P_{z1} + P_{z2}) - (C_{z1} + C_{z2})] > R - \gamma(I_f + I_L)$, the payoffs of receiving $P_z$ and the possibility of interaction $\gamma$ are improved for loners, compared with their initial values. Hence, the probability of receiving tends to 1 for loners. Therefore, the willingness to receive messages is enhanced as the payoffs of receiving and the possibility of interaction are improved. When the loners are willing to receive messages, their damage is reduced; thus, the ability to collaborate is enhanced in the multi-agent system.

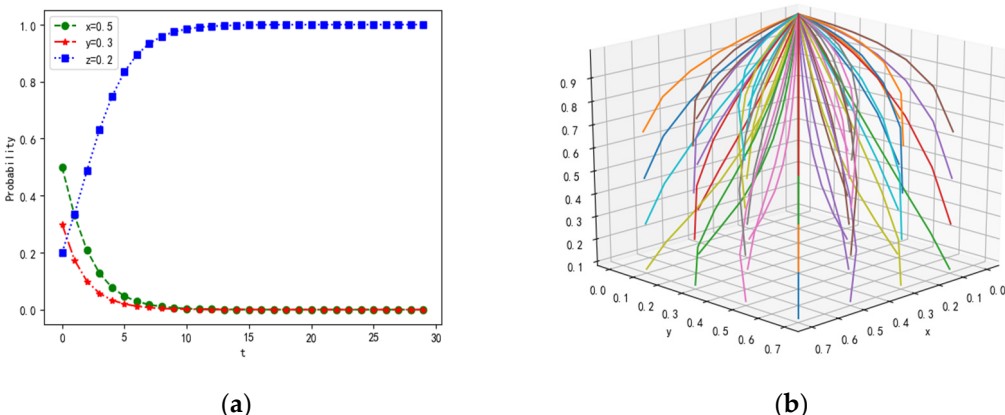

**Figure 3.** Evolutionary results of $p_2(0,0,1)$ shown under a corresponding constraint condition, (**a**) represents the convergence of probability with time, and (**b**) shows the results of dynamic simulation in three dimensions.

### 4.1.3. Scenario 3

In terms of the stability conditions of $p_3(1,0,0)$, the parameter values are assumed as follows: $\lambda = 0.5$, $\alpha = 0.5$, $\beta = 0.2$, $\gamma = 0.1$, $p = 0.3$, $R_f = 20$, $C_f = 5$, $P_{L1} = 15$, $P_{L2} = 10$, $C_{L1} = 3$, $C_{L2} = 1$, $R_L = 20$, $C_L = 5$, $P_{z1} = 15$, $P_{z2} = 15$, $C_{z1} = 1$, $C_{z2} = 1$, $I_f = 15$, $I_L = 15$, $R = 10$. The probability of strategies remains unchanged. The simulation results are shown in Figure 4.

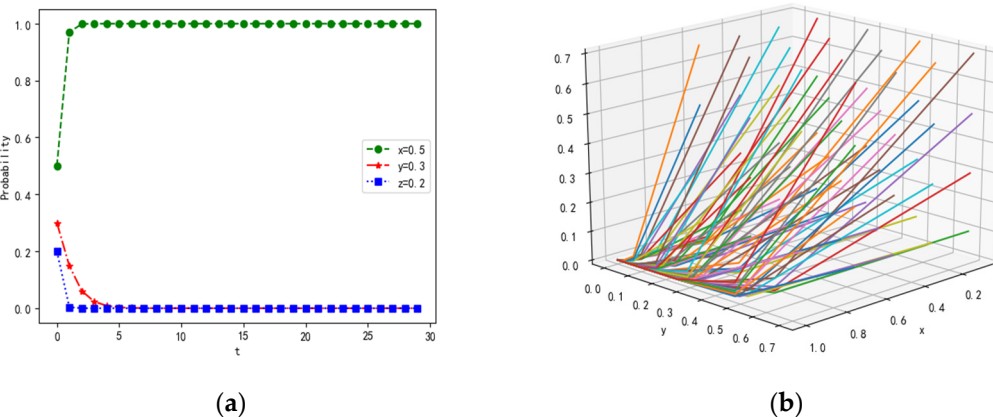

(a)          (b)

**Figure 4.** Evolutionary results of $p_3(1,0,0)$ shown under a corresponding constraint condition, (**a**) represents the convergence of probability with time, and (**b**) shows the results of dynamic simulation in three dimensions.

For the condition of $\alpha R_f - C_f > 0$, $p(P_{L1} - P_{L2}) + (C_{L2} - C_{L1}) < [\beta R_L - C_L](\lambda - 1)$ and $\lambda[\beta(P_{z1} + P_{z2}) - (C_{z1} + C_{z2})] < R - \gamma(I_f + I_L)$, the positivity of feedback $\alpha$ is improved for followers, compared with initial values. Hence, the probability of the strategies of feedback tends to 1 for followers. That is, the followers can track leaders and share information with each other in real-time, strengthening their ability to cooperate with each other in the multi-agent system. The possibility that agents are influenced by others with the power to destroy collaboration is reduced; thus, the ability to collaborate is enhanced in the system.

### 4.1.4. Scenario 4

On the basis of the stability conditions of $p_4(1,1,0)$, the parameters values are $\lambda = 0.5$, $\alpha = 0.5$, $\beta = 0.2$, $\gamma = 0.1$, $p = 0.5$, $R_f = 20$, $C_f = 5$, $P_{L1} = 25$, $P_{L2} = 15$, $C_{L1} = 3$, $C_{L2} = 1$, $R_L = 20$, $C_L = 5$, $P_{z1} = 15$, $P_{z2} = 15$, $C_{z1} = 1$, $C_{z2} = 1$, $I_f = 15$, $I_L = 15$, $R = 10$. The probability of the strategies remains unchanged. The simulation results are shown in Figure 5.

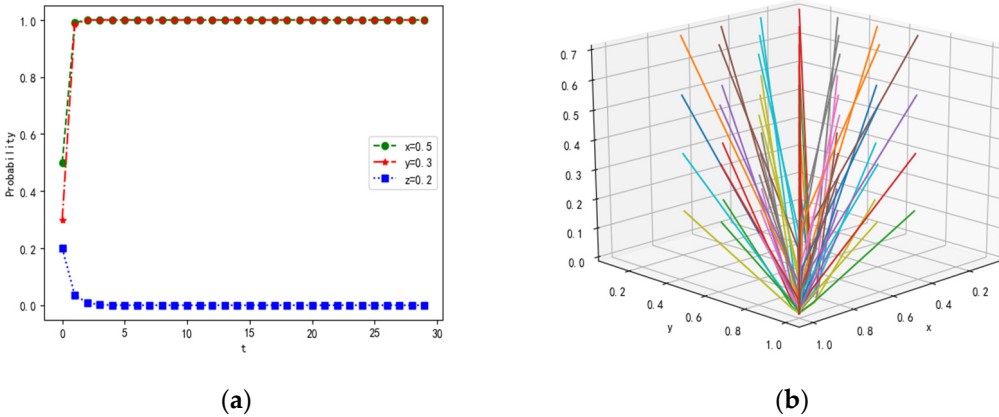

(a)          (b)

**Figure 5.** Evolutionary results of $p_4(1,1,0)$ shown under a corresponding constraint condition, (**a**) represents the convergence of probability with time, and (**b**) shows the results of dynamic simulation in three dimensions.

For the condition of $\alpha R_f - C_f > 0$, $p(P_{L1} - P_{L2}) + (C_{L2} - C_{L1}) > [\beta R_L - C_L](\lambda - 1)$ and $\lambda[\beta(P_{z1} + P_{z2}) - (C_{z1} + C_{z2})] < R - \gamma(I_f + I_L)$, the probability of successful sending $p$ and payoffs $P_{L1}, P_{L2}$ is improved for leaders, compared with the parameter values at point $p_4(1, 0, 0)$. Hence, the probability of the strategies to send also tends to 1; that is, sending accurate messages is a prerequisite of successful communication in the multi-agent system.

### 4.1.5. Scenario 5

According to the stability conditions of $p_5(1, 0, 1)$, the parameter values are set to $\lambda = 0.5$, $\alpha = 0.5$, $\beta = 0.2$, $\gamma = 0.3$, $p = 0.3$, $R_f = 20$, $C_f = 5$, $P_{L1} = 15$, $P_{L2} = 10$, $C_{L1} = 5$, $C_{L2} = 1$, $R_L = 15$, $C_L = 1$, $P_{z1} = 20$, $P_{z2} = 20$, $C_{z1} = 1$, $C_{z2} = 1$, $I_f = 15$, $I_L = 15$, $R = 5$. The probability of strategies remains unchanged. The simulation results are shown in Figure 6.

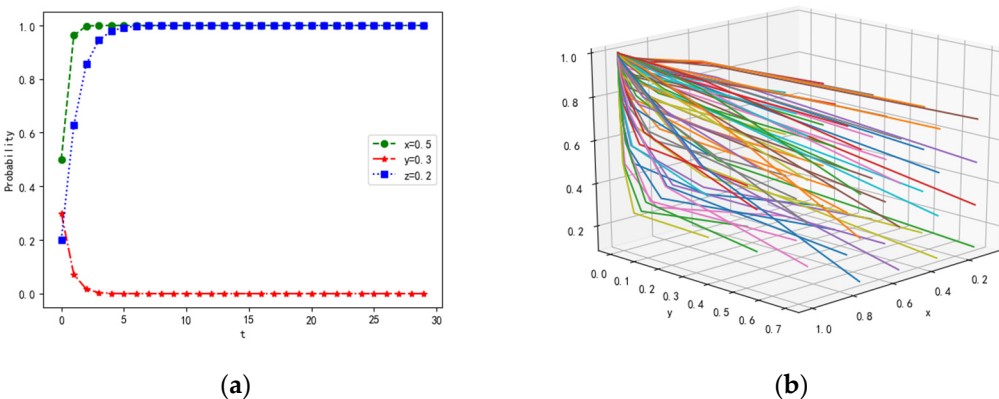

(a)  (b)

**Figure 6.** Evolutionary results of $p_5(1, 0, 1)$ shown under a corresponding constraint condition, (**a**) represents the convergence of probability with time, and (**b**) shows the results of dynamic simulation in three dimensions.

For the condition of $\alpha R_f - C_f > 0$, $p(P_{L1} - P_{L2}) + (C_{L2} - C_{L1}) < [\beta R_L - C_L](\lambda - 1)$ and $\lambda[\beta(P_{z1} + P_{z2}) - (C_{z1} + C_{z2})] > R - \gamma(I_f + I_L)$, the positivity of feedback $\alpha$ is improved for followers, and the payoffs of unreceiving $R$ decreases, compared with the parameters' values in $p_2(0, 0, 1)$. Hence, the probability of the strategies of feedback also tends to 1 for the followers; that is, the followers can track leaders and share information with each other in real time, strengthening the ability of agents to cooperate with each other. The damage of loners is reduced due to the decrease in $R$; thus, the ability to collaborate is enhanced in the multi-agent system. Figure 6 shows the difference in graphs.

### 4.1.6. Scenario 6

On the basis of the stability conditions of $p_6(0, 1, 0)$, the parameter values are $\lambda = 0.5$, $\alpha = 0.2$, $\beta = 0.2$, $\gamma = 0.1$, $p = 0.5$, $R_f = 10$, $C_f = 5$, $P_{L1} = 20$, $P_{L2} = 20$, $C_{L1} = 3$, $C_{L2} = 1$, $R_L = 20$, $C_L = 5$, $P_{z1} = 15$, $P_{z2} = 15$, $C_{z1} = 1$, $C_{z2} = 1$, $I_f = 15$, $I_L = 15$, $R = 10$. The probability of strategies remains unchanged. The simulation results are shown in Figure 7.

For the condition of $\alpha R_f - C_f < 0$, $p(P_{L1} - P_{L2}) + (C_{L2} - C_{L1}) < 0$ and $\lambda[\beta(P_{z1} + P_{z2}) - (C_{z1} + C_{z2})] < R - \gamma(I_f + I_L)$, the positivity of feedback $\alpha$ and rewards $R_f$ decreased for the followers, compared with the parameter values in point $p_5(1, 1, 0)$. Meanwhile, the payoffs of sending $P_{L1}$, $P_{L2}$ reduce for leaders. Hence, the probability of the strategies of feedback tends to 0. The convergence speed of the strategy of receiving declines. In this scenario, loners do not send messages on time, and the followers are inactive to feedback, resulting in delayed collaboration and tracking errors in the multi-agent system.

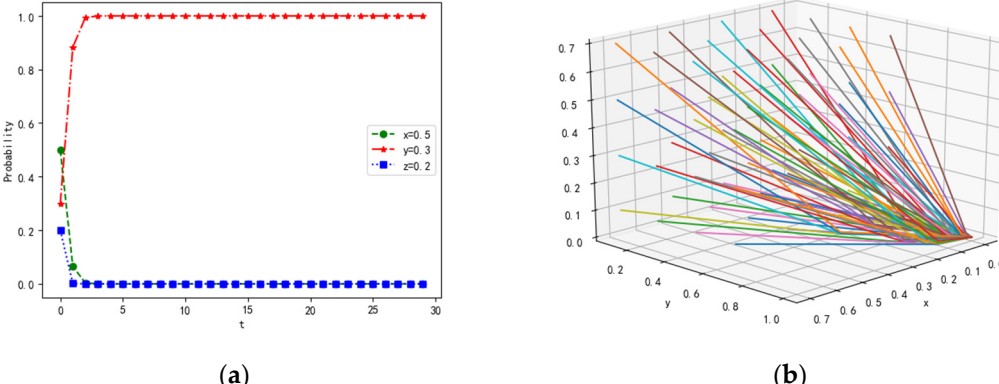

(a)                                                (b)

**Figure 7.** Evolutionary results of $p_6(0, 1, 0)$ shown under a corresponding constraint condition, (**a**) represents the convergence of probability with time, and (**b**) shows the results of dynamic simulation in three dimensions.

#### 4.1.7. Scenario 7

For the stability conditions of $p_7(0, 1, 1)$, the parameter values are $\lambda = 0.5$, $\alpha = 0.2$, $\beta = 0.2$, $\gamma = 0.1$, $p = 0.5$, $R_f = 10$, $C_f = 5$, $P_{L1} = 20$, $P_{L2} = 10$, $C_{L1} = 3$, $C_{L2} = 1$, $R_L = 20$, $C_L = 5$, $P_{z1} = 20$, $P_{z2} = 20$, $C_{z1} = 1$, $C_{z2} = 1$, $I_f = 15$, $I_L = 15$, $R = 10$. The probability of strategies remains unchanged. The simulation results are shown in Figure 8.

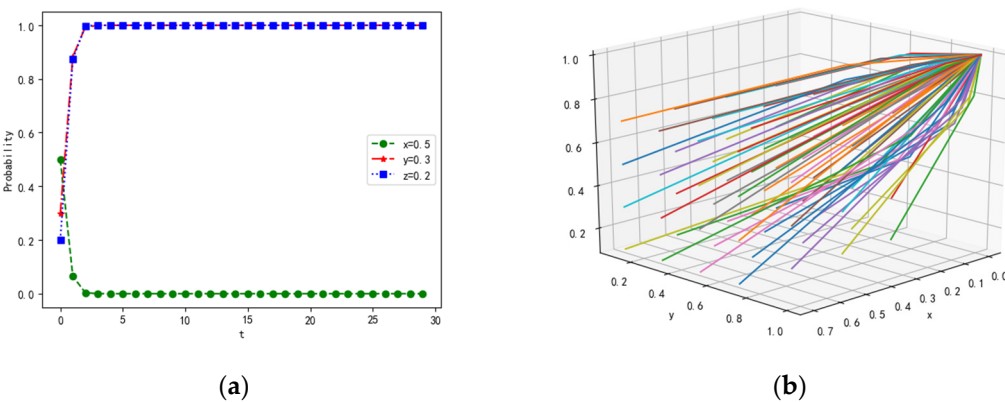

(a)                                                (b)

**Figure 8.** Evolutionary results of $p_7(0, 1, 1)$ shown a corresponding under a constraint condition, (**a**) represents the convergence of probability with time, and (**b**) shows the results of dynamic simulation in three dimensions.

For the condition of $\alpha R_f - C_f < 0$, $p(P_{L1} - P_{L2}) + (C_{L2} - C_{L1}) < 0$ and $\beta(P_{z1} + P_{z2}) - (C_{z1} + C_{z2}) > R - \gamma\left(I_f + I_L\right)$, the payoffs of receiving $P_{z1}$, $P_{z2}$ are improved for the loners, compared with the parameter values at point $p_6(0, 1, 0)$. Hence, the probability of the strategies of receiving also tends to 1. In this scenario, the loners can receive messages and promote the probability of interaction, reducing the destructive possibility of tracking to cooperate in the multi-agent system.

#### 4.1.8. Scenario 8

In terms of the stability conditions of $p_8(1, 1, 1)$, the parameter values are $\lambda = 0.5$, $\alpha = 0.5$, $\beta = 0.2$, $\gamma = 0.1$, $p = 0.5$, $R_f = 20$, $C_f = 5$, $P_{L1} = 20$, $P_{L2} = 10$, $C_{L1} = 3$, $C_{L2} = 1$, $R_L = 20$, $C_L = 5$, $P_{z1} = 20$, $P_{z2} = 20$, $C_{z1} = 1$, $C_{z2} = 1$, $I_f = 15$, $I_L = 15$, $R = 10$. The probability of strategies remains unchanged. The simulation results are shown in Figure 9.

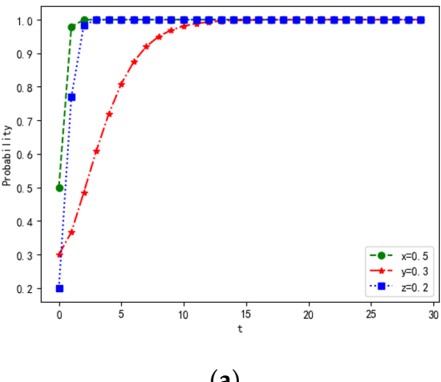
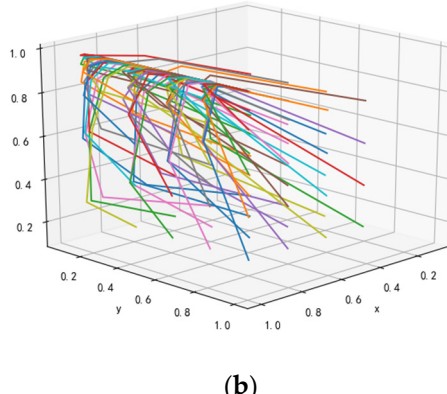

(a)            (b)

**Figure 9.** Evolutionary results of $p_8(1,1,1)$ shown under a corresponding constraint condition, (**a**) represents the convergence of probability with time, and (**b**) shows the results of dynamic simulation in three dimensions.

Under the conditions of $\alpha R_f - C_f > 0$, $p(P_{L1} - P_{L2}) + (C_{L2} - C_{L1}) > [\beta R_L - C_L](\lambda - 1)$ and $\beta(P_{z1} + P_{z2}) - (C_{z1} + C_{z2}) > R - \gamma(I_f + I_L)$, the payoffs of feedback $R_f$ and the positivity of feedback $\alpha$ are improved for the followers, compared with the parameter values at point $p_8(1,1,1)$. Hence, the probability of the strategies of feedback also tends to 1. In this scenario, the convergence speed of the loners decreases and that of the followers and leaders increases, enhancing the communication and cooperation capabilities of target tracking in the multi-agent system.

### 4.2. Impacts of Different Parameters on the Evolutionary Results

This analysis indicates that point $p_8(1,1,1)$ is an ideal ESS at eight equilibrium points. Though the initial values of the parameters do not affect the evolutionary results, they can affect the speed of convergence. Subsequently, we investigate the effect of parameters, such as the proportion of messages $\lambda$, the positivity of feedback $\alpha$ and receiving $\beta$, the possibility of interaction $\gamma$, and the probability of successful sending $p$ on mutual cooperation. The evolutionary results of $\lambda$, $\alpha$, $\beta$, $\gamma$, and $p$ are shown in the following section.

#### 4.2.1. Influence of Parameter $\lambda$ on Dynamic Evolution

Leaders send messages with a certain proportion, which can affect the accuracy of communication throughout the system. Sending all messages provides the basic guarantee for target tracking. On the contrary, lack of information leads to the deviation in tracking and affects the feedback of followers. Hence, exploring the effect of parameter $\lambda$ on followers, leaders, and loners is necessary. The other parameters are set as follows: $\alpha = 0.5$, $\beta = 0.2$, $\gamma = 0.1$, $p = 0.5$, $R_f = 20$, $C_f = 5$, $P_{L1} = 20$, $P_{L2} = 10$, $C_{L1} = 3$, $C_{L2} = 1$, $R_L = 20$, $C_L = 5$, $P_{z1} = 20$, $P_{z2} = 20$, $C_{z1} = 1$, $C_{z2} = 1$, $I_f = 15$, $I_L = 15$, $R = 10$. When $\lambda$ is 0.2, 0.5, and 0.8, the simulation results are as shown in Figure 10.

For the followers, as $\lambda$ increases, the probability of feedback remains unchanged, but the convergence speed increases with different proportions of feedback. Sending all messages can enhance leaders' performance to motivate the effectiveness of followers' feedback. With the increase in $\lambda$, the probability and convergence speed of sending are unchanged. $\lambda$ does not affect the probability and convergence speed of receiving of the loners.

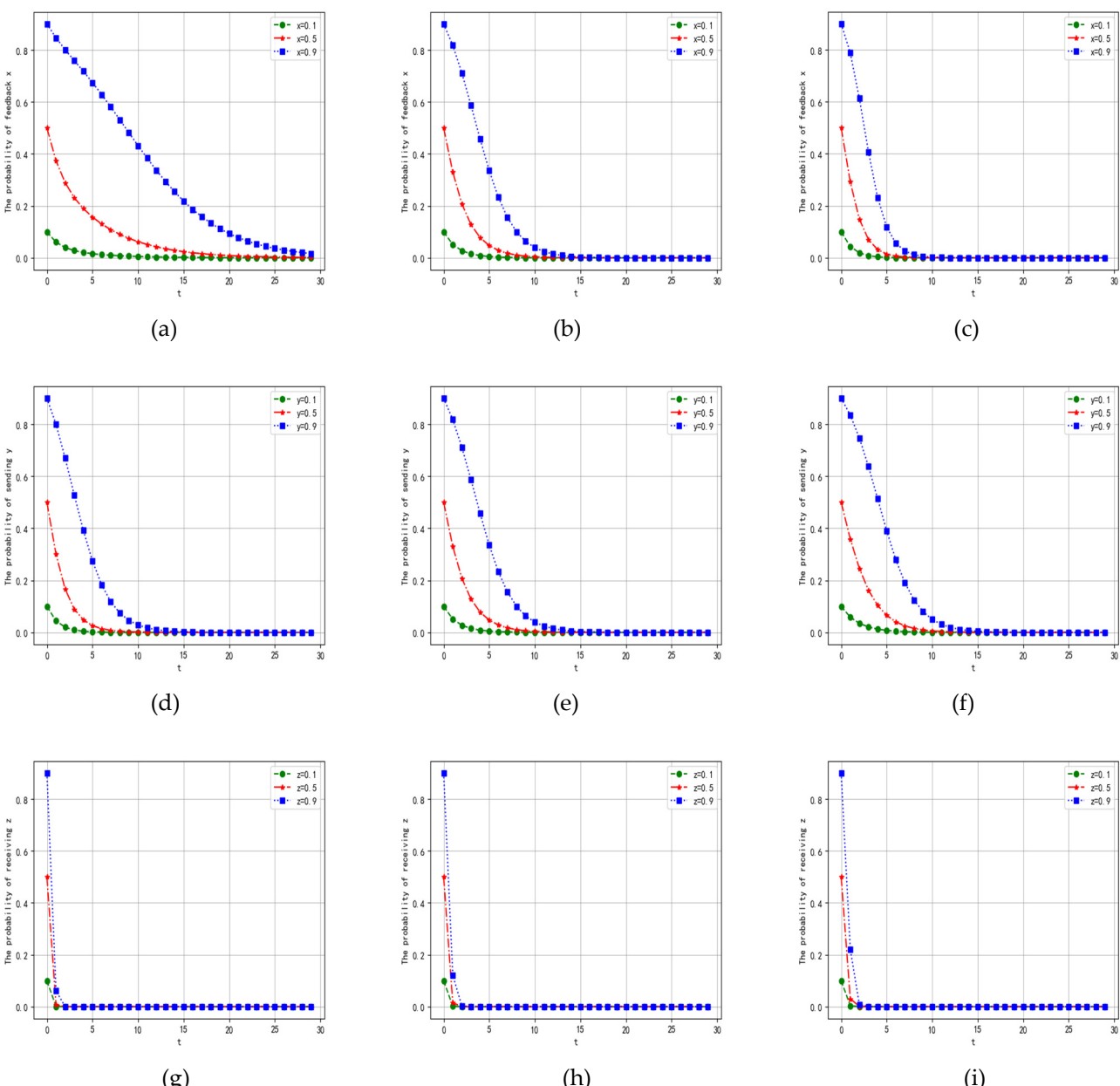

**Figure 10.** Evolutionary results of $\lambda$ shown under different probabilities of strategy. $\lambda$ is 0.2, 0.5, and 0.8, from left to right. (**a**–**c**) show that the probability of strategy $x$ changes when time $t$ is under the different conditions of $\lambda = 0.2$, 0.5, and 0.8. (**d**–**f**) show that the probability of strategy $y$ changes when time $t$ is under the different conditions of $\lambda = 0.2$, 0.5, and 0.8. (**g**–**i**) show that the probability of strategy $z$ changes when time $t$ is under the different conditions of $\lambda = 0.2$, 0.5, and 0.8.

4.2.2. Influence of Parameter $\alpha$ on Dynamic Evolution

Figure 11 elucidates the impact of parameter $\alpha$ on evolutionary results under different agents in the multi-agent system. The other parameters are assumed as follows: $\lambda = 0.5$, $\beta = 0.2$, $\gamma = 0.1$, $p = 0.5$, $R_f = 20$, $C_f = 5$, $P_{L1} = 20$, $P_{L2} = 10$, $C_{L1} = 3$, $C_{L2} = 1$, $R_L = 20$, $C_L = 5$, $P_{z1} = 20$, $P_{z2} = 20$, $C_{z1} = 1$, $C_{z2} = 1$, $I_f = 15$, $I_L = 15$, $R = 10$. The values of $\alpha$ are set to 0.2, 0.5, and 0.8, and the simulation results are shown in the following section.

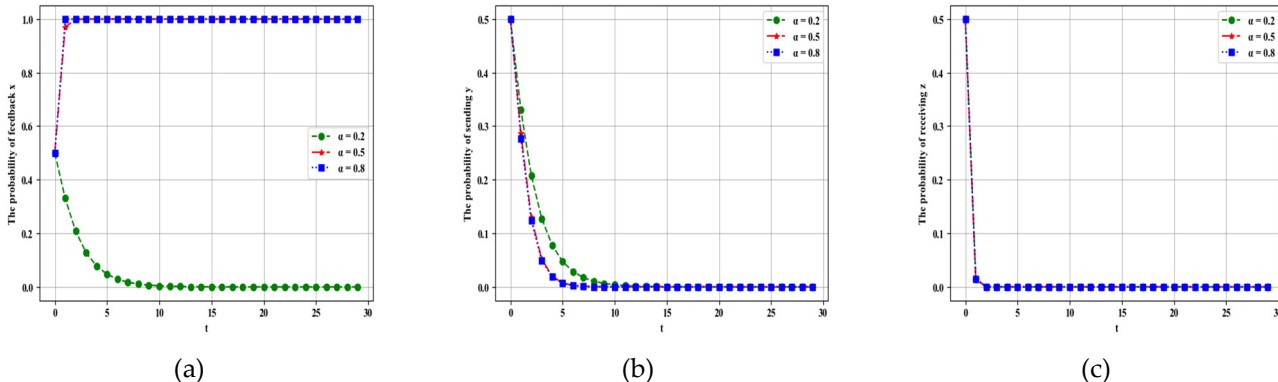

**Figure 11.** Evolutionary results of $\alpha$ shown under different probabilities of strategies $x$, $y$, and $z$ in (**a**), (**b**), and (**c**), respectively.

When the positivity of feedback $\alpha$ of the followers increases, the probability of feedback builds up from 0 to 1; thus, $\alpha$ can affect the selection of strategy of the followers. However, the probability of sending is not affected by the increase in $\alpha$ as it approaches 0. The convergence speed of sending is improved when the value of $\alpha$ increases. For the loners, with the increase in $\alpha$, the probability of receiving $z$ remains unchanged, and $z$ tends to 0 under different $\alpha$. Thus, the feedback's positivity evidently affects dynamic evolution.

### 4.2.3. Influence of Parameter $\beta$ on Dynamic Evolution

The positivity of receiving messages of a certain proportion can affect the accuracy of communication throughout the system. In fact, if messages are received by the leaders and loners, then this provides the basic guarantee for target tracking. On the contrary, lack of receiving information leads to deviation in tracking and affects the feedback for followers. Hence, exploring the effect of parameter $\beta$ on followers, leaders, and loners is necessary. Other parameters are $\lambda = 0.5$, $\alpha = 0.2$, $\gamma = 0.1$, $p = 0.5$, $R_f = 20$, $C_f = 5$, $P_{L1} = 20$, $P_{L2} = 10$, $C_{L1} = 3$, $C_{L2} = 1$, $R_L = 20$, $C_L = 5$, $P_{z1} = 20$, $P_{z2} = 20$, $C_{z1} = 1$, $C_{z2} = 1$, $I_f = 15$, $I_L = 15$, $R = 10$. When $\beta$ is 0.2, 0.5, and 0.8, the simulation results are as shown in Figure 12.

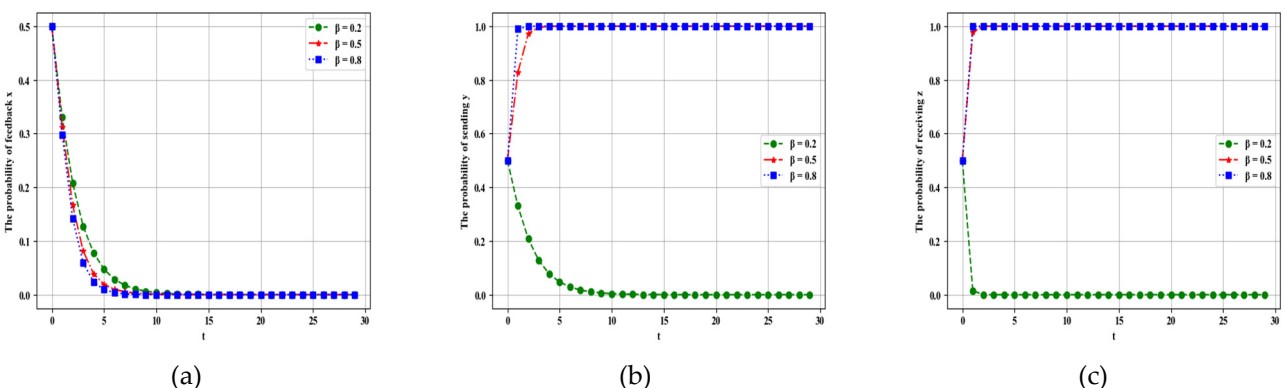

**Figure 12.** Evolutionary results of $\beta$ shown under different probabilities of strategies $x$, $y$, and $z$ in (**a**), (**b**), and (**c**), respectively.

As $\beta$ increases, the probability of feedback remains unchanged, but the probability of sending $y$ and receiving $z$ increases from 0 to 1. Receiving feedback messages and transmitting messages can enhance the system's performance to motivate the effectiveness of feedback, sending, and receiving. With the increase in $\beta$, the probability and the convergence speed of sending of the followers are unchanged. In terms of loners and leaders, $\beta$ does not affect the convergence and the speed of receiving and sending.

### 4.2.4. Influence of Parameter $\gamma$ on Dynamic Evolution

Figure 13 illustrates the impact of parameter $\gamma$ on the evolutionary results under different agents in the multi-agent system. Other parameters are assumed as follows: $\lambda = 0.5$, $\alpha = 0.5$, $\beta = 0.2$, $p = 0.5$, $R_f = 20$, $C_f = 5$, $P_{L1} = 20$, $P_{L2} = 10$, $C_{L1} = 3$, $C_{L2} = 1$, $R_L = 20$, $C_L = 5$, $P_{z1} = 20$, $P_{z2} = 20$, $C_{z1} = 1$, $C_{z2} = 1$, $I_f = 15$, $I_L = 15$, $R = 10$. The values of $\gamma$ are set to 0.2, 0.5, and 0.8, and the simulation results are shown in the following section.

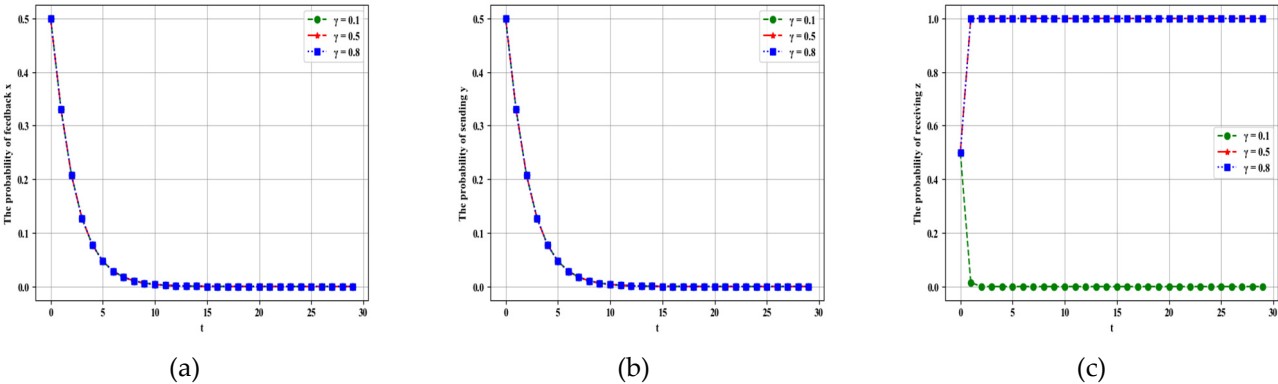

(a)　　　　　　　　　(b)　　　　　　　　　(c)

**Figure 13.** Evolutionary results of $\gamma$ shown under different probabilities of strategies $x$, $y$, and $z$ in (**a**), (**b**), and (**c**), respectively.

When the possibility of interaction $\gamma$ increases, the probability of feedback and sending remains unchanged. We found that $\gamma$ can affect the selection of strategy for loners. The probability of receiving increases from 0 to 1 with the increase in $\gamma$, and its convergence speed also increases. The dynamic evolution of collaboration is enhanced with the increase in interactive possibilities in multi-agent systems. In summary, the possibility of interaction $\gamma$ increases, thereby enhancing the positivity of receiving messages. Therefore, this condition is favorable when communicating with each other.

### 4.2.5. Influence of Parameter $p$ on Dynamic Evolution

The probability of successful sending with different proportions can affect the accuracy of communication throughout the system. Hence, exploring the effect of parameter $p$ on the system is necessary. Other parameters are $\lambda = 0.5$, $\alpha = 0.5$, $\beta = 0.2$, $\gamma = 0.1$, $R_f = 20$, $C_f = 5$, $P_{L1} = 20$, $P_{L2} = 10$, $C_{L1} = 3$, $C_{L2} = 1$, $R_L = 20$, $C_L = 5$, $P_{z1} = 20$, $P_{z2} = 20$, $C_{z1} = 1$, $C_{z2} = 1$, $I_f = 15$, $I_L = 15$, $R = 10$. When $p$ is 0.2, 0.5, and 0.8, the simulation results are shown in Figure 14.

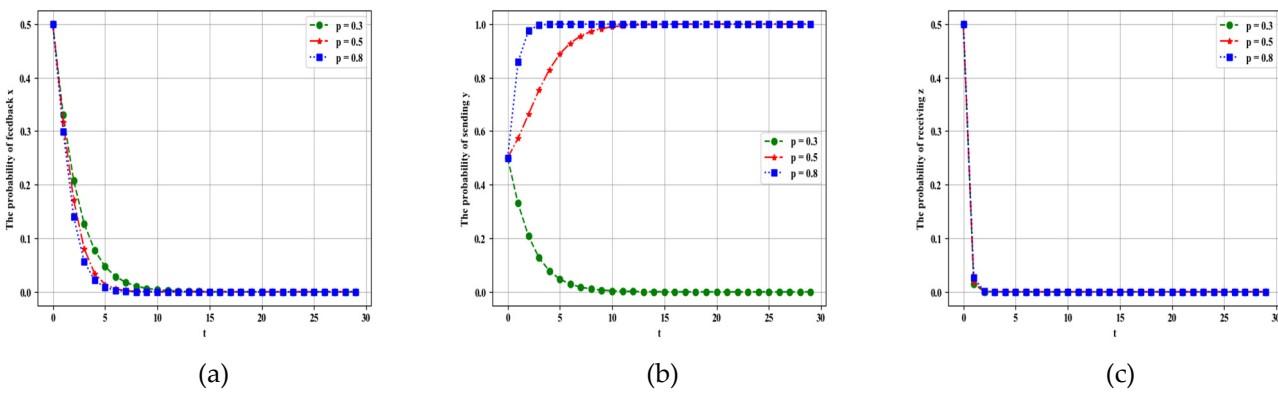

(a)　　　　　　　　　(b)　　　　　　　　　(c)

**Figure 14.** Evolutionary results of $p$ shown under different probabilities of strategies $x$, $y$, and $z$ in (**a**), (**b**), and (**c**), respectively.

With the increase in $p$, the probability of receiving $z$ remains unchanged, but the probability of sending $y$ increases from 0 to 1. The convergence speed of feedback and

sending is improved. In fact, if the messages are sent by leaders successfully, then the basic guarantee for target tracking is provided. On the contrary, lack of sending information leads to deviation in tracking and affects the feedback. As $p$ increases, the convergence speed of feedback increases for the followers. In terms of loners, $p$ does not affect the convergence speed of receiving. In summary, the probability of successful sending $p$ increases, thereby enhancing the positivity of feedback and sending messages to communicate with each other.

## 5. Conclusions and Policy Enlightenment

In summary, we have demonstrated the evolution of collaboration based on evolutionary games. This study initially develops the model of different strategies, different parameters, and interaction in a multi-agent system of followers, leaders, and loners. Subsequently, after setting up the replication dynamic equation of different roles, equilibrium points are obtained to confirm the constraint conditions of the evolutionary stable strategy. Then, the research focuses on the influence of strategies and parameters on the dynamic evolutionary results of collaboration in different scenarios. The simulation results indicate that to gain the optimal results, followers should feedback messages to leaders positively while receiving messages from the leaders and transmitting messages to the loners. The leaders send all messages to the followers and loners; the loners receive the messages from the followers and leaders. In fact, the results have shown that the leaders played an important role in the collaboration. If all messages are sent by the leaders successfully, then they provide a basic guarantee for the accurate exchange of information. On the contrary, the lack of sending all information leads to a deviation in tracking and an impact on feedback. In terms of a multi-agent system, collaboration is key to ensure that all agents harmoniously form a unified entirety in an expectant approach.

In addition, according to the simulation results, we found that the consistency of collaboration is in an optimal state when the stakeholders agree to achieve a common goal of exchanging information. That is, leaders send all messages, followers feedback messages to leaders on time, and loners receive messages positively, as shown in Figure 9. This result demonstrates that the effectiveness of our proposed model is reasonable.

This study elucidates some policy implementations of the model realized by the followers, leaders, and loners for collaboration in evolutionary games. The probability of strategies is affected by their obtained payoffs and costs in the system. Therefore, promoting their payoffs is necessary to enhance the positivity of interaction while decreasing costs of communication with each other. For the followers, the payoffs of receiving and transmitting are increased to motivate the positivity of feedback messages. Then, improving the rewards of feedback in the communication process between followers and leaders is reasonable. For the leaders, sending all messages is necessary to provide the basis of mutual communication. On the contrary, if partial messages are sent, then integral communication is affected in the multi-agent system. Meanwhile, reducing the costs of sending can enhance interaction with agents. The probability of the successful sending of messages should be improved to provide a basic guarantee for collaboration, which can ensure that all agents receive messages. For the loners, increasing the possibility of interaction with followers and leaders is the most important to reduce hindrance to the system. That is, as much interaction as possible between loners and others is a good decision, thereby ensuring that all agents harmoniously form a unified entirety in an expectant approach. Then, through cooperation between agents, the basic capabilities of each agent are improved, and their social behavior can be further understood from the interaction of the agents. In summary, in a dynamic and open environment, agents with different goals must coordinate their goals and resources. During conflicts between resources and goals, if the coordination of agents fails to reach a better situation, then a deadlock occurs. This condition causes the agents to be unable to carry out their next step of work. On the contrary, if all agents can reach an agreement of collaboration in a multi-agent system, then the exchange of information is enhanced to improve cooperation.

**Author Contributions:** Z.G.: methodology, visualization, and writing of the original draft. Y.D.: review. Z.G. and Y.D.: preliminary investigations. All authors have read and agreed to the published version of the manuscript.

**Funding:** This research was funded by the National Science Foundation (62073270, 61673016, 61703353), Innovation Research Team of the Education Department of Sichuan province (15TD0050), and the Graduate Innovative Research Project of Southwest Minzu University, Project number: CX2020SZ99.

**Acknowledgments:** We really appreciate the support from the National Science Foundation (62073270, 61673016, 61703353), Innovation Research Team of the Education Department of Sichuan province (15TD0050), and the Graduate Innovative Research Project of Southwest Minzu University, Project number: CX2020SZ99.

**Conflicts of Interest:** The authors declare no conflict of interest.

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
