# Peer review of "Dynamic Model of Collaboration in Multi-Agent System Based on Evolutionary Game Theory"

_games, doi:10.3390/g12040075_

Round 1
Reviewer 1 Report
The manuscript carries a good idea in the field of evolutionary game theory and fits well the aim and scope of the journal. This manuscript explores how a specific feature of the multi-agent system, mutual collaboration, could be treated with the evolutionary game theory.
- Manuscript requires moderate English changes;
- Abstract : reviewer suggest to revise the abstract within the 200 word limit;
- Introduction:
- Line 59 : The sentence "One of the articles [29] manifests the tracking 59 controller model of a leader-followers formation based on actuated underwater vehicles (UUVs) in three dimensional 60 space." is not clear. The expression refers to the general context of the literature background or to the specific authors of the reference number 29 ? . Reviewer suggests to re-write sentence;
- Revise the english language for this section.
- Section 2
- Sub- section 2.1 : Table 1 caption should be located in the next page of manuscript;
- Sub- section 2.2: Table 2 caption and the first part of the table itself should be located in the next page of manuscript;
- Section 3:
- Section 3 heading schould be located in the next page of the manuscript;
- Sub-section 3.1 heading should be removed and re-written is as preamble of Section 3;
- Theorem 1 and proof of the Theorem 1 must be better formatted;
- Table 3 should be located in the next page of manuscript;
- All Section 4 heading should be better located;
-
Conclusion must be improved particularly the Policy enlightenment sub-section
- The Policy enlightenment sub-section should be incorporated in the main conclusion section
- The Conclusions sub-section heading should be removed
Author Response
Dear Editors and Reviewers,
Thank you for your letter and for the reviewers' comments concerning our manuscript entitled “The Dynamics Model of Mutual Collaboration in Multi-agent System Based on Evolutionary Game Theory” (ID: games-1328178). Those comments are all valuable and very helpful for revising and improving our paper, as well as the important guiding significance to our researches. We have studied comments carefully and have made correction which we hope meet with approval. Revised portion was marked by highlighting in the paper, and comments were made in the corresponding position. In the abstract, abstract, introduction and conclusions, these parts have been carefully modified. In addition to this, the tables, formulas and subtitles in the article have all been modified.
We appreciate for Editors/Reviewers’ warm work earnestly, and hope that the correction will meet with approval. Once again, thank you very much for your comments and suggestions.
The main corrections in the paper and the responds to the reviewer's comments areas following:
Responds to the reviewer's comments:
Reviewer#1:
- Response to comment: (Abstract: reviewer suggest to revise the abstract within the 200 word limit;)
Response: We are very sorry for our incorrect writing. We have made correction according to the Reviewer’s comments. In abstract, we cut out the extra parts to limit words within 200.
- Response to comment: (Introduction: Line 59 : The sentence "One of the articles [29] manifests the tracking 59 controller model of a leader-followers formation based on actuated underwater vehicles (UUVs) in three dimensional 60 space." is not clear. The expression refers to the general context of the literature background or to the specific authors of the reference number 29? Reviewer suggests to re-write sentence; Revise the english language for this section.)
Response: We have re-written this part according to the Reviewer’s suggestion. For the sentence of line 59, they are modified as “One of the articles [29] manifests the problem of formation control based on leader-followers model in three dimensional space, which explores the persistent excitation of the desired formation to achieve the exponential stabilization of actual formation in terms of shape and scale”.
- Response to comment: (Section 2. Sub- section 2.1: Table 1 caption should be located in the next page of manuscript; Sub- section 2.2: Table 2 caption and the first part of the table itself should be located in the next page of manuscript.)
Response: We are very sorry for our incorrect writing. We have made correction according to the Reviewer’s comments. We have modified the caption of Table 1 and Table 2 to locate in the next page of manuscript.
- Response to comment: (Section 3: Section 3 heading should be located in the next page of the manuscript; Sub-section 3.1 heading should be removed and re-written is as preamble of Section 3; Theorem 1 and proof of the Theorem 1 must be better formatted; Table 3 should be located in the next page of manuscript.)
Response: Considering the reviewer’s suggestion, we have deleted the subtitles of Sub-section 3.1. Section 3 heading and Table 3 has been located in the next page of the manuscript. We have formatted Theorem 1 and proof of the Theorem.
- Response to comment: (All Section 4 heading should be better located; Conclusion must be improved particularly. The Policy enlightenment sub-section should be incorporated in the main conclusion section. The Conclusions sub-section heading should be removed.)
Response: Considering the reviewer’s suggestion, we have re-written this part according to the Reviewer’s suggestion. We have deleted the subtitles of conclusions and incorporated policy enlightenment sub-section in conclusions. Conclusions have been improved particularly. And all section 4 heading have been better located.
Finally, the language of the full text has been carefully revised.
Special thanks to you for your good comments.

Reviewer 2 Report
I appreciate your effort to improve your initial submission.
However, I think that the paper is still hard to read and the language still needs considerable improvement from that perspective. The text is often misleading and hardly understandable. As ist is, it will put off potential readers.
A list of issues (it is by far not comprehensive):
The very first sentence: "Mutual collaboration of multi-agent is..." Collaboration is mutual by definition and you are talking about agents in multi agent systems. I do not know what a "multi-agent" is. Is it agents in a MAS?
Use abbreviations like MAS for multi agent system and make clear earn you talk about a single agent and when about the whole system.
Layout of payoff matrix (line 188): Is there a third column? Please, improve the layout.
line 193: "Letrepresent the E_x expected ..." should be "Let E_x represent the expected value...."
....
line 294: "the successful probability of sending". Is it the "probability of successfully sending"?
line 443: "With the increase of beta...". Should it not be p?
....
Equation (12), (16) and (20): Do you take derivatives of the average expected values? Then it should be x(E_x - E1-x) (in 12 and analogously in (16) and (20)) which would correspond to the result.
Author Response
Dear Editors and Reviewers,
Thank you for your letter and for the reviewers' comments concerning our manuscript entitled “The Dynamics Model of Mutual Collaboration in Multi-agent System Based on Evolutionary Game Theory” (ID: games-1328178). Those comments are all valuable and very helpful for revising and improving our paper, as well as the important guiding significance to our researches. We have studied comments carefully and have made correction which we hope meet with approval. Revised portion was marked by highlighting in the paper, and comments were made in the corresponding position. In the abstract, abstract, introduction and conclusions, these parts have been carefully modified. In addition to this, the tables, formulas and subtitles in the article have all been modified.
We appreciate for Editors/Reviewers’ warm work earnestly, and hope that the correction will meet with approval. Once again, thank you very much for your comments and suggestions.
The main corrections in the paper and the responds to the reviewer's comments areas following:
Responds to the reviewer's comments:
Reviewer#2:
- Response to comment: (A list of issues (it is by far not comprehensive): The very first sentence: "Mutual collaboration of multi-agent is..." Collaboration is mutual by definition and you are talking about agents in multi agent systems. I do not know what a "multi-agent" is. Is it agents in a MAS? Use abbreviations like MAS for multi agent system and make clear earn you talk about a single agent and when about the whole system.)
Response: We are very sorry for our incorrect writing. We have made correction according to the Reviewer’s comments. We have changed “multi-agent” to “agent”. MAS is multi-agent system.
- Response to comment: (Layout of payoff matrix (line 188): Is there a third column? Please, improve the layout.)
Response: We have re-written this part according to the Reviewer’s suggestion. For layout of payoff matrix (line 188), there is not a third column. We have improved the layout.
- Response to comment: (line 193: "Let represent the Ex expected ..." should be "Let Ex represent the expected value....")
Response: We are very sorry for our incorrect writing. We have made correction according to the Reviewer’s comments. We have modified these sentences, such as “Let Ex represent the expected value....”
- Response to comment: (line 294: "the successful probability of sending". Is it the "probability of successfully sending"?)
Response: Considering the reviewer’s suggestion, we have modified this sentence as “the probability of successfully sending”.
- Response to comment: (line 443: "With the increase of beta...". Should it not be p?)
Response: Considering the reviewer’s suggestion, we have re-written this part according to the Reviewer’s suggestion. We have modified all this sentence as “with the increase of p” in this paper.
- Response to comment: (Equation (12), (16) and (20): Do you take derivatives of the average expected values? Then it should be x (Ex - E1-x) (in 12 and analogously in (16) and (20)) which would correspond to the result.)
Response: Considering the reviewer’s suggestion, according to the definition of the dynamic equation of replication, we take F(x)=dx/dt =x(Ux –U1x)=x(1-x)(Ux-U1-x)=……,which would correspond to the result.
Finally, the language of the full text has been carefully revised.
Special thanks to you for your good comments.

Round 2
Reviewer 1 Report
After author/s implementation, the present manuscript suits well the aim and the scope of the journal.
Author Response
Dear Editors and Reviewers,
Thank you for your letter and for the reviewers' comments concerning our manuscript entitled “The Dynamics Model of Mutual Collaboration in Multi-agent System Based on Evolutionary Game Theory” (ID: games-1328178). Those comments are all valuable and very helpful for revising and improving our paper, as well as the important guiding significance to our researches. We have studied comments carefully and have made correction which we hope meet with approval. Revised portion was marked by highlighting in the paper, and comments were made in the corresponding position. In the abstract, introduction, model, simulation and conclusions, these parts have been carefully modified. In addition to this, the tables, formulas and subtitles in the article have all been modified.
We appreciate for Editors/Reviewers’ warm work earnestly, and hope that the correction will meet with approval. Once again, thank you very much for your comments and suggestions.

Reviewer 2 Report
The paper becomes acceptable, though the language could still be improved.
The first sentence of the abstract ist still unclear.
Collaboration of multi-agent is of great significance to reduce the frequency of error in message communication and enhance the consistency of exchanging information.
Can you first just state the problem and its significance?
Author Response
Dear Editors and Reviewers,
Thank you for your letter and for the reviewers' comments concerning our manuscript entitled “The Dynamics Model of Mutual Collaboration in Multi-agent System Based on Evolutionary Game Theory” (ID: games-1328178). Those comments are all valuable and very helpful for revising and improving our paper, as well as the important guiding significance to our researches. We have studied comments carefully and have made correction which we hope meet with approval. Revised portion was marked by highlighting in the paper, and comments were made in the corresponding position. In the abstract, introduction, model, simulation and conclusions, these parts have been carefully modified. In addition to this, the tables, formulas and subtitles in the article have all been modified.
We appreciate for Editors/Reviewers’ warm work earnestly, and hope that the correction will meet with approval. Once again, thank you very much for your comments and suggestions.
The main corrections in the paper and the responds to the reviewer's comments areas following:
Responds to the reviewer's comments:
Reviewer#1:
- Response to comment: (Can you first just state the problem and its significance? )
Response: We are very sorry for our incorrect writing. We have made correction according to the Reviewer’s comments. In the first sentence of the abstract, multi-agent collaboration is greatly important to reduce the frequency of error in message communication and enhance the consistency of exchanging information. This study explored the process of evolutionary decision and stable strategies among multi-agent system, including followers, leaders, and loners, involved in collaboration based on evolutionary game theory (EGT).

This manuscript is a resubmission of an earlier submission. The following is a list of the peer review reports and author responses from that submission.
Round 1
Reviewer 1 Report
Language issues make reading the paper very hard and there is a high risk that statements will be misunderstood (e.g. line 97: multi-agent is an important -> system?, is missing several times; Line 203: As a result, gaining ? 1 − ? 8 is y equilibrium points of system model). The second example appears in the proof of theorem 1. I frankly to not understand it.
A major concern is the at least confusing and not quite consistent mathematical notation (e.g. indexing of x in 2.3, derivation of dynamic equation must be explained in more detail).
What instruments were used for the simulations performed?
The paper needs extensive improvement of the language so that statements are clear. Mathematical notation of costs and payoffs should be revised.
Author Response
I am very grateful to the reviewer for his careful review of the article, I have carefully revised the comments given by reviewers.
Firstly, after modification, the exact expression should be that Multi-Agent System is an important branch of distributed artificial intelligence, and several independent agents are adopted to achieve common goals in this system. These agents have the autonomous ability to coordinate with each other.
Second, in system, when F(x)=0, F(y)=0,F(z)=0, eight local equilibrium points of three-species adopting pure strategies can be obtained: p1(0,0,0), p2(0,0,1), p3(1,0,0), p4(1,1,0), p5(1,0,1), p6(0,1,0), p7(0,1,1), p8(1,1,1). Proof of proposition: When x=0 or x=1,y=0 or y=1,z=0 or z=1, permanent establishment F(x)=0,F(y)=0,F(z)=0, so p1-p8 are the equilibrium point of the system.
Then, for the derivation of equations, we first figure out the payoffs of different strategies. According to the theory of replication dynamic equations, payoffs are brought into expected payoffs, then the average expected payoffs can be acquired by calculation.
In addition, the simulation results are obtained by MATLAB and python tools to implement.
Finally, the paper's abstract, introduction and other parts have extensive improvement of the language so that statements are clear. Mathematical notation of costs and payoffs have been revised.
Reviewer 2 Report
The paper addresses the problem of consistency in multi-agent systems. A model is defined, including followers, leaders and loners, and subsequently analyzed using evolutionary game theory. Simulation experiments are carried out to reach conclusions.
Although the structure of the paper is clear, and while reading the conclusion, I had more or less an idea what the paper was aiming at, this is unclear in the abstract and introduction. Basic elements that need to be present were vaguely described or expecting readers to be familiar with the background literature already. I give some example questions which needs to be addressed properly and especially in the abstract and introduction:
- What is the problem this paper addresses? I read the problem of consistency. But more explanation is necessary at this point.
- What is the purpose/aims of the paper? Both scientifically and societal, in the conclusion the authors talk about policies, but nothing has been said about this in the Introduction.
- How will the paper achieve those aims? design a model, run experiments, or both. Substantiate your research design.
- Which background literature is the paper building forth on? The introduction starts with a 'dive' into the literature, but since the reader is not given any context this is hard to follow. I suggest to first introduce the points mentioned before and make background literature a separate section.
- Explain the concepts introduced, what is a high dimensional multi-agent system?
When I started reading section 2 it became more clear what the paper was aiming for. The model introduced seems basic, what is its high dimensional property which you claim to contribute to?
The eight evolutionary stable strategies (EES) result in eight scenarios that are simulated, but again explanation to the reader is missing. An explanation of each scenario, also making clear what distinguishes them would improve the paper a lot.
How the results of the simulations relate to the consistency problem? I do not see that anywhere discussed.
Scenario 4 concludes by saying that sending accurate messages is a prerequisite of successful communication. Isn't that an obvious conclusion?
In general I doubt the contribution of the conclusions "mutual collaboration is key ...". What is new about this result? Is it the fact that 3 agents are included in the model? Or the fact that EGT is used? Or the simulation after determining the ESS? I cannot find an answer to these questions in the paper.
I value the fact that the authors have thought about policy implications, however, this part could improve by giving a context in which this could be applied and make precise which results a policy maker could use or create new insights.
The use of English could improve, I suggest to have it checked by a native speaker.
Author Response
I am very grateful to the reviewer for his careful review of the article, I have revised the abstract and introduction of the paper strictly according to the comments of the reviewers. And now I reply to review comments one by one.
Firstly, the question that this paper is going to address is to reduce the frequency of error in message communication and enhance the consistency of exchanging information. Meanwhile, in introduction, new content has been added to explain consistency in more detail in the first paragraph.
Second, the purpose/aims of the paper is that through the cooperation between agents, the basic capabilities of each agent are not only improved, but also the social behavior can be further understood from the interaction of agents. In a dynamic and open multi-agent environment, multi-agents with different goals must coordinate their goals and resources. When there is a conflict between resources, if the coordination of multi-agents fails to reach a better situation, a deadlock will occur. This will cause agents to be unable to carry out their next step of work.
Then, the paper achieves those aims by following process. This study explored the process of the evolutionary decision and stable strategies among multi-agent system, including followers, leaders and loners, involved in mutual collaboration of multi-agent system based on evolutionary game theory (EGT). The main elements that affected the strategies of multi-agent were discussed and the three-dimensional evolution model is established. The evolutionary stability strategy (ESS) and stable conditions were analyzed later. Through MATLAB simulation, numerical simulation results were obtained and they manifested that leaders play an important role in exchanging information with other agents, accepting agents’ state information and sending messages to agents. Then, with the positivity of receiving and feeding back messages for followers, it is profitable for system to implement message communication, and a high positivity of agents can accelerate exchange of information. In the early system of mutual collaboration, two controllers of leaders and followers were prevailing. Different from the previous works, we introduce loners into system to simulate the impact of environmental factors on mutual collaboration. Under this circumstance, leaders send messages to followers and loners, and after receiving information, they feed back messages to leaders. At the behavior level, reducing costs can help to strengthen the punishment of impeding the exchange of information and improve the positivity of mutual collaboration, so as to facilitate the evolutionary converge towards the ideal state. Finally, EGT results reveal that the possibility of mutual collaboration between loners and others is improved, as well as the rewards are increased, which can promote the implementation of message communication that encourages leaders to send all messages, improves the feedback positivity of followers and reduces the hindering degree of loners.
In addtion, we supplement some literature for further explanation. A high dimensional multi-agent system is the system of three stakeholders, which was established by replicating the dynamic equations in three dimensions. Meanwhile, the contributions of this paper are presented in three-dimensions spaces as: Different from the previous game model involved only two stakeholders of leadership-follower, this model investigate the influence factors and the exchange of information among three stakeholders effectively. We introduce loners into system to simulate the impact of environmental factors on mutual collaboration. Results reveal that the possibility of mutual collaboration between loners and others is improved, as well as the rewards are increased, which can promote the implementation of message communication that encourages leaders to send all messages, improves the feedback positivity of followers and reduces the hindering degree of loners.
Five, the main distinguishes of evolutionary stable strategies (EES) results in eight scenarios are changing payoffs, costs and degree parameters under different stability conditions, which can affect the probability of strategies' choice, and ultimately affect mutual collaboration.
Six, the simulations are related to the consistency problem. According to simulation results, we found that the consistency of mutual collaboration is optimal state when thee stakeholders come to an agreement to achieve a common goal of mutual exchange information. Namely, leaders send all messages, followers feed back messages to leaders in time, as well as loners receive messages positively, which has been shown in figure 9. This result demonstrates that the effectiveness of our proposed model is reasonable.
Finally, this study explored the process of the evolutionary decision and stable strategies among multi-agent system, including followers, leaders and loners, involved in mutual collaboration of multi-agent system based on evolutionary game theory (EGT). The main elements that affected the strategies of multi-agent were discussed and the three-dimensional evolution model is established. The evolutionary stability strategy (ESS) and stable conditions were analyzed later.
Reviewer 3 Report
The manuscript carries a good idea in the field of evolutionary game theory and fits well the aim and scope of the journal. This manuscript explores how a specific feature of the multi-agent system, mutual collaboration, could be treated with the evolutionary game theory.
- It is necessary an English language, style, and text-editing revisions;
- It is necessary to explain in a more appropriate way the description of the model (Section 2) since it is described not clearly;
- The reviewer suggests to re-write and implement the Introduction due to the fact that in some part of it it seems to be created through the cut and paste procedure;
- Author/s should consider the possibility to make some adjustments in the arrangement of tables and figures (Section 2 and Section 3);
- Author/s should take into consideration the necessity to revise the layout of Table 1 in Section 2;
- It is necessary to revise the preamble of Section 2 before describing the following sub-sections. The reviewer suggests explaining in which way the section is structured describing in a few words the contents of the following sub-sections.
- Author/s might consider the idea to re-write the abstract complied to the journal author's guidelines and in particular, taking into consideration the number of words and better explanation of the question addressed in a broad context highlighting in a more clear way the purpose of the study;
- Lines 13-14: The sentence "Through investigating the dynamic model of different agents' behavior and their evolutionary stable strategy (ESS) that is to be in line with the replicator dynamic equations, EGT enables to make a quantitative analysis of consistency and interactive." is not clear. Author/s should consider re-write the sentence underlying what EGT acronymous means.
Author Response
I am very grateful to the reviewer for his careful review of the article, I have revised the abstract and introduction of the paper strictly according to the comments of the reviewers. And now I reply to review comments one by one.
Firstly, through the relevant literature research and reference, the expression of model has been improved. And we revise and supplement details in the part of introduction and abstract.
Second, we have made improvements to the tables and graphs. For the layout of Table 1 in Section 2, we revise the description of parameters.
Then, we have revised the preamble of Section 2, in this section, the dynamics model of mutual collaboration based on evolutionary game theory involving each stakeholder is proposed. And then, payoffs matrix of agents are obtained according to the parameters of agents’ behaviors. In addition, the tripartite replication dynamic equation is derived.
Finally, we have modified most of the content, especially the abstract and introduction. EGT is the abbreviation of evolutionary game theory.